# Macrosphelide A Exhibits a Specific Anti-Cancer Effect by Simultaneously Inactivating ENO1, ALDOA, and FH

**DOI:** 10.3390/ph14101060

**Published:** 2021-10-19

**Authors:** Kyoung Song, Nirmal Rajasekaran, Chaithanya Chelakkot, Hun Seok Lee, Seung-Mann Paek, Hobin Yang, Lina Jia, Hee Geon Park, Woo Sung Son, Yu-Jin Kim, Joon-Seok Choi, Hae Min Jeong, Young-Ger Suh, Hwayoung Yun, Young Kee Shin

**Affiliations:** 1College of Pharmacy, Duksung Women’s University, Seoul 01369, Korea; songseoul17@duksung.ac.kr; 2Laboratory of Molecular Pathology and Cancer Genomics, Research Institute of Pharmaceutical Sciences and College of Pharmacy, Seoul National University, Seoul 08826, Korea; nirmalpharma@gmail.com (N.R.); ryanlee0109@gmail.com (H.S.L.); dbdyd99@naver.com (H.Y.); 3Bio-MAX Institute, Seoul National University, Seoul 08826, Korea; cgchaithanyalakshmi@gmail.com; 4College of Pharmacy and Research Institute of Pharmaceutical Sciences, Gyeongsang National University, Jinju 52828, Korea; million@gnu.ac.kr; 5Department of Pharmacology, Shenyang Pharmaceutical University, Shenyang 110016, China; frankjln@syphu.edu.cn; 6Department of Molecular Medicine and Biopharmaceutical Sciences, Graduate School of Convergence Science and Technology, Seoul National University, Seoul 08826, Korea; pong1418@snu.ac.kr; 7Department of Pharmacy, College of Pharmacy and Institute of Pharmaceutical Sciences, CHA University, Pocheon-si 13496, Korea; wson@cha.ac.kr (W.S.S.); ygsuh@cha.ac.kr (Y.-G.S.); 8Laboratory of Cancer Genomics and Molecular Pathology, Samsung Medical Center, Sungkyunkwan University School of Medicine, Seoul 03063, Korea; bubble@skku.edu; 9College of Pharmacy, Daegu Catholic University, Hayang-ro 13-13, Gyeongsan-si 38430, Korea; joonschoi@naver.com; 10Genopharm Inc., Seoul 08394, Korea; Hm.jeong@genopharmbio.com; 11College of Pharmacy, Pusan National University, Busan 46241, Korea

**Keywords:** macrosphelide A, Warburg effect, ENO1, ALDOA, FH

## Abstract

Aerobic glycolysis in cancer cells, also known as the Warburg effect, is an indispensable hallmark of cancer. This metabolic adaptation of cancer cells makes them remarkably different from normal cells; thus, inhibiting aerobic glycolysis is an attractive strategy to specifically target tumor cells while sparing normal cells. Macrosphelide A (MSPA), an organic small molecule, is a potential lead compound for the design of anti-cancer drugs. However, its role in modulating cancer metabolism remains poorly understood. MSPA target proteins were screened using mass spectrometry proteomics combined with affinity chromatography. Direct and specific interactions of MSPA with its candidate target proteins were confirmed by in vitro binding assays, competition assays, and simulation modeling. The siRNA-based knockdown of MSPA target proteins indirectly confirmed the cytotoxic effect of MSPA in HepG2 and MCF-7 cancer cells. In addition, we showed that MSPA treatment in the HEPG2 cell line significantly reduced glucose consumption and lactate release. MSPA also inhibited cancer cell proliferation and induced apoptosis by inhibiting critical enzymes involved in the Warburg effect: aldolase A (*ALDOA*), enolase 1 (*ENO1*), and fumarate hydratase (*FH*). Among these enzymes, the purified ENO1 inhibitory potency of MSPA was further confirmed to demonstrate the direct inhibition of enzyme activity to exclude indirect/secondary factors. In summary, MSPA exhibits anti-cancer effects by simultaneously targeting *ENO1*, *ALDOA*, and *FH*.

## 1. Introduction

In most cancer cells, metabolic reprogramming causes atypical metabolism, characterized by increased total glycolysis, lactate production, and defective mitochondrial ATP production in hypoxic and normal oxygen conditions [1,2,3,4]. This phenomenon, known as the Warburg effect or aerobic glycolysis, was first described by Otto Warburg in the 1920s [5]. Although the reasons why aggressive cancer cells prefer to use an inefficient method of burning glucose remain a puzzle, this metabolic adaptation is believed to be critical for a malignant phenotype. The reprogramming of cancer metabolism is induced by genetic dysregulation, involving various oncogenes and tumor suppressors such as *RAS*, *AKT*, *MYC*, *PI3K*, *mTOR*, *TP53*, and *PTEN*, resulting in cancer cell proliferation and invasion [6,7]. Accumulating evidence indicates that reprogrammed cancer metabolism is a hallmark of cancer cells that could be potential therapeutic targets [4]. 

Altered energy metabolism in cancer cells is accompanied by the upregulation of several glycolytic enzymes, such as *GLUT1*, hexokinase 2, phosphofructokinase, enolase 1 (*ENO1*), aldolase A (*ALDOA*), lactate dehydrogenase A, and pyruvate dehydrogenase kinase, which stimulate the Warburg effect [4]. These glucose-metabolizing enzymes critically regulate a variety of cellular processes [8]. Many extensive studies have shown that inhibiting glycolytic enzymes disrupts the Warburg effect or cell proliferation, making them a fascinating target for anti-cancer therapy. 

Natural products are attractive as resources for new anti-cancer drug development. Macrosphelide A (MSPA), a derivative of a natural product [9], belongs to a group of macrolactone polyketides that have been the source of several drug leads due to their unique biological activities and structural features. Previously, we reported a method for use in the concise syntheses of MSPA and the discovery of a more cytotoxic MSPA derivative [10,11,12]. MSPA shows anti-cancer properties in a variety of cell lines, and the compound inhibits the major features of carcinogenesis. Several mechanisms, including apoptotic effects and the inhibition of cell–cell adhesion, are proposed to explain the anti-cancer effects of MSPA [5,13,14]; however, the cellular targets of MSPA have not yet been investigated. 

Here, we studied the molecular mechanisms that underlie the anti-cancer actions of MSPA. We conducted mass spectrometry coupled with affinity chromatography to identify the putative targets for MSPA as *ENO1*, *ALDOA*, and fumarate hydratase (*FH*). MSPA strongly binds to *ENO1*, *ALDOA*, and *FH* and reduces their enzymatic activity. The loss of enzymatic activity impaired the proliferation of cancer cells, glucose consumption, and lactate release, suggesting that MSPA targets the Warburg effect. Our results identify a new anti-cancer mechanism of MSPA and endorse the potential therapeutic relevance of MSPA as an anti-cancer drug. 

## 2. Results 

### 2.1. Synthesis of MSPA and Its Derivatives

The successful synthesis of (+)-MSPA and several other compounds was accomplished in our previous study (Appendix A) [11,15]. MSPA–biotin chimeras were synthesized for use in a competitive binding assay and an in vitro binding assay (Appendix A). The biotinylated linker on MSPA the increased structural flexibility and the sensitivity of detection. Because linker length can be critical for identifying biological target proteins, we synthesized two MSPA–biotin chimeras, one containing a medium-length linker (MSPA-735) and one containing a long-length linker (MSPA-961). Macrosphelide B, which is structurally similar to MSPA except at the C-14 position (MSPA: OH; MSPB: O), was synthesized as a control to validate novel MSPA targets (Appendix A, left panel). Butenylated biotin, a linker-conjugated biotin structure, was also synthesized as a negative control for biotin-tagged MSPA because some biotin chemicals and linker compounds directly bind proteins (Appendix A, right panel). Our previous study achieved the successful synthesis of (+)-MSPA and several related compounds by the efficient preparation and assembly of key fragments, including the 16-membered ring formation from highly labile precursors [11].

### 2.2. Selective Cytotoxic Activity of MSPA on Cancer Cells 

The cytotoxic effect of MSPA was evaluated in several cancer cell lines, namely HepG2, HL60, and MCF-7, and non-cancer cell lines THLE-3, peripheral blood mononuclear cells (PBMC), and MCF-10A. Each cell line was treated with 12.5 μM of MSPA, and the number of viable cells was analyzed for up to 96 h after MSPA treatment using an ATP measurement method (Figure 1A). There was a significant difference in the cytotoxic effect of MSPA in cancer and normal cells. The viability of the cancer cell lines HepG2, HL60, and MCF-7 was 51.7%, 54.6%, and 48.4%, respectively, while the viability of the non-cancer cell lines THLE-3, PBMC, and MCF-10 was 78.2%, 86.2%, and 94.5%, respectively. MSPA had a more potent and selective cytotoxic effect against cancer cells than against normal cells. To identify the mechanism of the MSPA-induced cytotoxic effect on cancer cells, HepG2 and HL60 cells were treated with 12.5 μM of MSPA for 72 h and analyzed via flow cytometry after double staining the cells with Annexin V/7-amino-actinomycin D (7-AAD) (Figure 1B). Annexin V binds the phosphatidylserine (PS) on the outer leaflet of the early apoptotic cell membrane, whereas 7-AAD binds DNA and is efficiently excluded by cells with an intact membrane. Thus, 7-AAD staining indicates a loss of integrity of the membrane in late apoptotic or necrotic cells. The HL-60 cell dot plot showed positive staining with only Annexin V-FITC (27.8%), whereas the HepG2 cell dot plot showed double staining with Annexin V and 7-AAD (14.1%), indicating that cells had undergone early and late-stage apoptosis after MSPA treatment. Staining with 7-ADD did not reveal any evidence of necrotic cell death. The effect of MSPA (500 µM) on apoptosis in HepG2 cells was also confirmed via immunofluorescence imaging using the apoptosis indicator Annexin V Alexa Fluor 488 (Figure 1C). These findings suggest that MSPA significantly increases cytotoxicity by mediating apoptosis in cancer cells.

### 2.3. Identification of MSPA Target Molecules

To investigate the putative targets that mediate the apoptotic effects of MSPA in cancer cells, we conducted a mass spectrometry proteomic screen coupled with affinity chromatography. Biotinylated MSPA, synthesized as an MSPA–biotin chimera, was used to identify MSPA-binding proteins (Appendix A). Biotinylated MSPA was incubated with HL60 cell lysates for 24 h, and then proteins captured with biotinylated MSPA were precipitated by incubation with streptavidin beads. MSPA-binding proteins were separated via sodium dodecyl sulfate–polyacrylamide gel electrophoresis (SDS-PAGE). Approximately ten distinct protein bands were detected by Coomassie brilliant blue staining (Figure 2A). Among them, two major bands were analyzed via mass spectrometry (Table 1). The proteins with the highest coverage levels were enzymes involved in glycolysis, including *ENO1*, *FH*, enolase 2 (*ENO2*), *GAPDH*, *ALDOA*, transaldolase 1 (*TALDO1*), and aldolase C (*ALDOC*). We performed immunoprecipitation assays to validate MSPA target protein interactions. We selected four putative targets, *ALDOA*, *GAPDH*, *ENO1*, and *FH*, for further validation based on their ranking from the analysis of mass spectrometry data. Those selected targets were confirmed to bind to MSPA specifically. MSPA could bind to the target proteins regardless of linker length or biotinylation (Figure 2B). We also investigated whether MSPA directly bound *ALDOA*, *GAPDH*, *ENO1*, and *FH* using in vitro binding assays, in which each of the purified four target proteins were incubated with biotinylated MSPA. Two additional substances were used as negative controls, butenylated biotin and MSPB, the latter of which is structurally similar to MSPA (Appendix A). We found that biotin-conjugated MSPA bound directly to each of the target proteins, but biotin-conjugated MSPB did not interact with any of the targets, suggesting that the binding between MSPA and the target proteins is highly specific to the MSPA structure. These specific binding interactions were further confirmed using a competitive binding assay. We added excess non-biotinylated MSPA at concentrations of 12.5 µM, 125 µM, and 1.25 mM as a competitor of biotinylated MSPA to HL60 cell lysates (Figure 2D). The binding of biotinylated MSPA to target proteins decreased in a concentration-dependent manner in the presence of non-biotinylated MSPA, suggesting that MSPA binding was highly specific. Thus, MSPA could bind directly and specifically to target proteins, including *ALDOA*, *GAPDH*, *ENO1*, and *FH*.

### 2.4. MSPA Inhibits Target-Enzyme Activities 

To investigate the effect of MSPA on the enzymatic activity of the identified target protein, HepG2 cells were treated with 100 µM of MSPA, and the enzymatic activity levels of *ALDOA*, *GAPDH*, *ENO1*, and *FH* were measured using corresponding in vitro enzyme assays (Figure 3A). We found that compared with the control (enzymatic activity set at a value of 1), MSPA binding significantly decreased the enzymatic activities of *ENO1* (0.72), *ALDOA* (0.78), and *FH* (0.80) while it had no effect on the enzyme activity of GAPDH. In order to exclude the possibility that the enzyme activity inhibition could be the effect of indirect/secondary factors occurring in the experiment and not by MSPA, we further confirmed the inhibitory effect of MSPA on purified ENO1 among MSPA target enzymes (Appendix A). MSPA showed the significant inhibition of purified enolase enzyme detected as a measure of PEP generated from 2PGA. Previous studies have shown that the pre-incubation of an inhibitor with enzyme significantly improves the inhibitory effect as opposed to adding the substrate along with the inhibitor [16,17]. Here, we pre-incubated MSPA with the purified ENO1 for 16 h before adding 2-phosphoglycerate to check whether it would have significance on enolase inhibition via MSPA. The pre-incubation of MSPA showed a more significant and profound inhibition of ENO1 activity than when MSPA was added along with the substrate. These results demonstrated that ENO1 inhibition via MSPA could be direct effects. Next, we confirmed MSPA’s inhibitory effect on glycolysis by using lactate production and glucose consumption as surrogate markers. As shown in Figure 3B,C, the amount of lactate and glucose in the culture media of MSPA-treated HepG2 cells was significantly lower than in control cells (NT). In addition, the effect of MSPA treatment on real-time cellular respiration was determined via the Seahorse energy phenotype assay, which measured the OCR and ECAR of cells at baseline and in response to the addition of stressor mixture, consisting of oligomycin, an ATP synthase inhibitor, and FCCP, a proton uncoupling agent. Oligomycin addition blocks ATP production via oxidative phosphorylation and FCCP addition helps probe the maximal respiration rate. Consistent with our previous observations, MSPA treatment significantly reduced the extracellular acidification rate (ECAR) and the oxygen consumption rate (OCR) in HepG2 cells, at both the basal level and upon treatment with stressor mixture (Figure 3D,E). These results indicate that MSPA can inhibit glycolytic as well as mitochondrial respiration.

To further investigate the inhibitory effect of MSPA on ENO1, ALDOA, and FH enzyme activity, HepG2, and MCF-7 cells were treated with enzyme-specific siRNAs. The silencing efficiency of each target siRNA was confirmed by qRT-PCR measurement of mRNA and Western blot analysis of protein expression levels (Figure 4A,B). Similar to the cytotoxic effects of MSPA treatment, all enzyme-specific-siRNA treatments had significant anti-proliferative effects (Figure 4C; *p* < 0.0001 for *ALDOA*, *ENO1*, and FH). siRNA against *ENO1* produced the most significant anti proliferative effect (44.8 ± 3.7%), followed by siRNAs against ALDOA and FH, which were 56.8 ± 10.8% and 59.3 ± 2.3%, respectively. Treatment with siRNAs (32.8 ± 3.6%) against *ALDOA* and *ENO1* together resulted in significantly more inhibition of cell proliferation than siRNA against *ALDOA* (56.8 ± 10.8%) or *ENO1* (44.8 ± 3.7%) alone. 

### 2.5. Molecular Docking and Molecular Dynamics Simulation

In silico experiments revealed the binding pattern between MSPA and its target proteins, ENO1, ALDOA, and FH. Molecular docking calculations and molecular dynamics simulations revealed that MSPA preferentially binds to *ENO1* at the molecular level (Figure 5). 

From the binding energy pattern of MSPA–enzyme complexes obtained for 50 ns in aqueous solution, the MSPA–*ENO1* complex showed higher energy values than the other complexes, indicating that the MSPA–*ENO1* complex was more stable than the other complexes (Figure 5A and Appendix A). The binding site of the MSPA–*ENO1* complex was stabilized with hydrogen bonds, salt bridges, and hydrophobic interactions, and there was a larger number of these interactions in the MSPA–*ENO1* complex than in other MSPA–enzyme complexes (Figure 5B and Appendix A). The results of these in silico experiments indicate that MSPA can act more effectively on *ENO1* than on either *ALDOA*, *ENO1*, or *FH* at the molecular level. These results reflected the result of the inhibition of ENO1 via MSPA in an enzyme assay but did not reflect the binding pattern of ALDOA and FH, which showed an inhibitory effect by MSPA, like ENO1. The presented in silico result was purely a prediction result using calculation. The experimental verification of the predicted results would be more certain with biophysical assay experiments with a high-purity enzyme at the molecular level.

## 3. Discussion

The deregulation of enzymes involved in glycolysis and the TCA cycle is frequently observed in several cancer types, and the inhibition of these enzymes is proving to be a promising strategy for cancer treatment [18,19,20]. This study shows that MSPA can simultaneously inhibit glycolysis and the TCA cycle in cancer cells, thereby inhibiting glucose metabolism and inducing cell death in cancer cells. 

MSPA, a derivative of natural product, could specifically bind to target proteins, including *ALDOA*, *ENO1*, and *FH,* which play a pivotal role in glycolysis and the TCA cycle. The MSPA binding effect with those enzymes showed various anti-cancer effects by decreasing each the activity of each enzyme.

MSPA exhibits selective cytotoxicity of cancer cells but not normal cells. To verify whether *ALDOA*, *ENO1*, and *FH* mediated MSPA-induced cytotoxicity, enzyme-specific siRNAs were treated in HepG2 cells, which showed a cytotoxic effect. These data suggest that *ALDOA*, *ENO1*, and *FH* targeting by MSPA could decrease cancer cell growth, mainly by simultaneously inactivating glycolysis and the TCA cycle. Interestingly, MSPA-induced cell death was observed after three days of treatment in cancer cells (Figure 1A). This late manifestation of cell death could be due to the high molecular size and low lipophilicity of MSPA. Further studies on the modification of the molecules could help improve the cell permeability and molecular effects of MSPA [12].

We further demonstrated more direct effects of MSPA on glycolysis and the TCA cycle. MSPA treatment in cancer cells significantly lowered cellular glucose consumption and lactate production, and a real-time metabolic phenotype assay (Seahorse assay) revealed that both the ECAR and OCR are attenuated (Figure 3). Binding partner analysis revealed that MSPA could form complexes with key glucose metabolism enzymes ENO1, ALDOA, and FH (Figure 5), and reduce their enzyme activity (Figure 3A), confirming that the simultaneous inhibition of glycolysis and the TCA cycle be the plausible mechanism of MSPA-induced cell death. Though aerobic glycolysis is a major source of energy in most cancer cells, the inhibition of glycolysis alone may not be sufficient for complete cell death as a population of fast-adapting cancer cells may reprogram their metabolic phenotype towards oxidative phosphorylation for survival [21]. Henceforth, the dual inhibition of glycolysis and the TCA cycle makes MSPA an attractive pharmacological target for anti-cancer drug development. Several studies have highlighted the importance of targeting glycolytic and TCA cycle enzymes in anti-cancer therapy [20,22,23].

Studies have identified many moonlighting functions for glycolytic enzymes apart from their role in glycolysis, escalating the need for the identification of potent inhibitors. As a result, many novel potent inhibitors of glycolytic enzymes have been identified and their anti-cancer properties elucidated [8,16,23,24]. Many of these enzymes, including GLUT1 [25,26], hexokinase II [27], GAPDH [28,29], lactate dehydrogenase [22], etc., are attractive targets for anti-cancer drug development and are currently in clinical or pre-clinical investigations [19]. In our study, we observed that MSPA forms most the stable complex with ENO1 (Figure 5B, Appendix A), and can significantly inhibit the enzyme activity of purified ENO1 (Appendix A). ENO1 overexpression is observed in several cancer types [30,31,32,33], and is known to be associated with cancer cell stemness [34]. The inhibition of ENO1 attenuates cell proliferation and induces cell death via glycolytic inhibition [16,24]; however, it is not unlikely that the cytotoxic effect observed is a combined effect of metabolic and non-metabolic functions of ENO1, including plasminogen activation, cell–cell adhesion, and DNA replication [35]. Moonlighting functions of ALDOA have also been demonstrated in a recent study by Gizak et al. A slow-binding inhibitor of ALDOA, UM0112176, induces cell death in cancer cells not via the inhibition of the enzymatic activity but through cytoskeletal disruption and caspase activation [8]. Similarly, FH has been implicated in non-metabolic functions, including response to DNA damage [36,37]. Though we demonstrate that MSPA significantly affects the enzymatic activity of ENO1, ALDOA, and FH, further studies are essential to demonstrate the role of MSPA in dysregulating the non-metabolic functions of these enzymes and to determine whether it contributes to the cytotoxicity observed. Furthermore, in vivo xenograft studies are imperative to fully demonstrate the anti-tumor effect of MSPA and to demonstrate the pharmacologic and clinical significance of MSPA as a drug target. In addition, the synergistic effects upon combination treatment with conventional chemotherapeutic and immune-modulating drugs also need to be investigated. 

## 4. Materials and Methods 

### 4.1. Cell Culture 

HL60 (human promyelocytic leukemia), HepG2 (human hepatocellular carcinoma), and MCF-7 (human breast adenocarcinoma) cells were purchased from the Korean Cell Line Bank. Peripheral blood mononuclear cells were obtained from Stem cell Technologies (Vancouver, BC, Canada) and grown in AIM-V assay medium (Invitrogen, Waltham, MA, USA). THLE-3 (human liver epithelial cells) and MCF-10A (human breast epithelial cells) were purchased from the American Type Culture Collection. All cell lines were cultured according to the manufacturer’s recommendations. Cells were routinely checked for mycoplasma contamination and authenticated using short tandem repeat DNA technology.

### 4.2. Cytotoxicity Assay

The number of viable cells was determined by quantifying ATP using a CellTiter Glo luminescent assay kit (Promega, Madison, WI, USA). Cells were seeded in a 96-well plate with 2000 cells per well, incubated for 24 h, and then treated with 12.5 μM of MSPA for 3–4 days. The luminescence was read using a GENios reader (Tecan Group Ltd., Zürich, Switzerland) with a set integration time. Live cells were stained with trypan blue and counted using a hemocytometer. 

### 4.3. Apoptosis Assay and Immunofluorescence

The apoptosis of cells treated with 500 μM of MSPA for 72 h was evaluated using annexin V staining. Cells treated with MSPA or the control (dimethyl sulfoxide) were stained using an annexin V– phycoerythrin (PE) apoptosis detection kit (BD Biosciences, San Jose, CA, USA) in accordance with the manufacturer’s protocol. Flow cytometry was performed using a FACS Calibur (BD Biosciences, San Jose, CA, USA). The gating was based on unstained control cells and annexin V–PE-stained and 7-amino-actinomycin D (7-AAD)-stained control cells. Briefly, cells grown in chambered slides were treated with either MSPA or the vehicle for 72 h and then washed once with ice-cold PBS, followed by two washes with binding buffer. Next, cells were incubated with fluorescein isothiocyanate-conjugated annexin V fluorescent antibody for 15 min, followed by a gentle wash with washing buffer. The cells were maintained in a binding buffer, and images were observed using a TE2000-U inverted fluorescence microscope (Nikon, Melville, New York, NY, USA) with JP4 filters (Chroma, Bellows Falls, VT, USA).

### 4.4. Mass Spectrometry

The identification of proteins in the bands of interest observed on SDS-PAGE gels was conducted via liquid chromatography/tandem mass spectrometry (LCMS/MS). The peptides were analyzed using LTQ ion trap MS (Thermo Fisher Scientific, Waltham, MA, USA). In brief, the gel bands were excised from the colloidal Coomassie blue-stained gels and proteolyzed with trypsin (Promega, Madison, WI, USA). The peptides were fractionated by reverse-phase high-performance liquid chromatography with an emitter column size of 10 μm using a 0–60% acetonitrile/0.5% formic acid gradient with a flowrate of 300 nL/min over 30 min. Peptide sequences were identified using Mascot (www.matrixscience.com, accessed on 1 January 2012) or Sequest (https://proteomicsresource.washington.edu/protocols06/sequest.php, accessed on 1 January 2012) software to search the National Center for Biotechnology Information non-redundant database with the acquired fragmentation data. Identified sequences were confirmed by manually inspecting the fragmentation spectra.

### 4.5. Immunoblotting, Immunoprecipitation, and In Vitro Binding Assay

#### 4.5.1. Immunoblotting

For immunoblotting experiments, protein samples from the total cell lysates were subjected to SDS-PAGE on 10% or 12% acrylamide/bisacrylamide gels. The protein profiles were electroblotted to PVDF (polyvinylidene difluoride) membranes (Merck Millipore, Burlington, MA, USA) at room temperature for two hours. The antibody anti-*ALDOA* (Cell Signaling Technologies, Danvers, MA, USA), anti-GAPDH (Cell Signaling Technologies, Danvers, MA, USA), anti-ENO1 (Cell Signaling Technologies, Danvers, MA, USA), and anti-FH (Cell Signaling Technologies, Danvers, MA, USA) incubation procedures and blocking buffer composition were conducted as per the supplier’s instructions.

#### 4.5.2. Immunoprecipitation

For immunoprecipitation experiments, the total cell lysate from cells incubated with butenylated biotin, biotinylated MSPA-961, or biotinylated MSPA-735 and then incubated with specific antibody overnight at 4 °C with a constant rotation, followed by the pull-down of the immune complex with streptavidin–agarose beads (Invitrogen, Waltham, MA, USA). The immunoprecipitated proteins were subjected to 10% SDS-PAGE and transferred onto PVDF membranes. Membranes were blocked in 5% non-fat milk in Tris-buffered saline with 0.1% Tween (TBS-T) buffer for 60 min. Primary antibodies against *ALDOA*, *GAPDH*, *ENO1*, and *FH* (Cell Signaling Technology, Danvers, MA, USA) were used individually at 1:1000 dilutions in 5% non-fat milk in TBS-T buffer for 90 min, and then washed with TBS-T buffer. Membranes were incubated with anti-rabbit secondary antibody (1:4000) in 5% non-fat milk in TBS-T buffer for 30 min and then washed in TBS-T buffer for 30 min. The bands were visualized using horseradish peroxidase-conjugated secondary antibodies (Thermo Fisher Scientific, Waltham, MA, USA).

#### 4.5.3. In Vitro Binding Assay

For the in vitro binding assay, biotinylated MSPA-961, biotin-MSPB, and butenylated biotin were synthesized [11] and then incubated with purified target proteins (*ALDOA*, *GAPDH*, *ENO1*, and *FH*). The mixture was incubated with the respective antibody to immunoprecipitate the target protein, and the co-immunoprecipitation of biotinylated MSPA-961 was examined using streptavidin–horseradish peroxidase. Butenylated biotin and biotin-MSPB served as negative controls.

### 4.6. Enzyme Activity Assay

The enzyme activity of *ENO1* in HepG2 cells treated with 100 µM of MSPA for 24 h was determined using an *ENO1* activity assay kit (Abcam, Cambrige, UK), and *FH* activity was assayed using a fumarase-specific activity microplate assay kit (Abcam, Cambridge, UK) according to the manufacturers’ instructions. *ALDOA* was measured using an aldolase activity assay kit (Biovision, Milpitas, CA, USA), and *GAPDH* was measured using a *GAPDH* activity assay kit (Biovision, Milpitas, CA, USA). HepG2 cells (1 × 10^6^) were treated with 100 µM of MSPA for 24 h and then rapidly homogenized in 100 µL of ice-cold aldolase/*GAPDH* assay buffer, and kept on ice for 10 min before the assays were performed according to the manufacturer’s instructions.

### 4.7. Enolase Activity Assay

Enolase inhibition via MSPA was measured using purified human ENO1. Enolase activity was measured directly by the appearance of phosphoenolpyruvate from 2-phosphoglycerate via absorption at 240 nm. The assay was conducted in 10 mM KCl, 5 mM MgSO_4_, and 100 mM triethanolamine at pH 7.4. For the pre-incubation assay, MSPA was preincubated with purified ENO1 for 16 h before the addition of 5 mM 2-phosphoglycerate. The assay was conducted in 96-well UV transmissible plates, and the absorbance was read at 240 nm using a Multiskan SkyHigh microplate spectrophotometer (Thermo Fischer Scientific, Waltham, MA, USA).

### 4.8. Glucose Consumption and Lactate Production

HepG2 cells were treated with different concentrations of MSPA (50 uM, 100 uM, 500 uM) for 72 h. Following incubation, the cell culture medium was analyzed for glucose consumption using the Glucose Assay Kit (#GAGO20; Sigma Aldrich). The lactate levels in the culture medium were determined using a Lactate Assay Kit (#MAK064; Sigma Aldrich), all in accordance with the manufacturer’s protocols. A cell-free medium was used as the blank control.

### 4.9. Seahorse Energy Phenotype Assay

The effect of MSPA treatment on the energy phenotype of HepG2 cells was determined using the Seahorse energy phenotype kit (Agilent Technologies, Santa Clara, CA, USA) following the manufacturer’s protocol. HepG2 cells (1 × 10^4^) were seeded in Seahorse 96-well cell culture microplates (Agilent technologies, Santa Clara, CA, USA) followed by MSPA treatment at the concentrations indicated. The Seahorse assay was conducted 48 h after MSPA treatment. On the day prior to the assay, sensor cartridges were hydrated in sterile water in a 37 °C non-CO2 incubator. After overnight hydration, the cartridges were calibrated in a calibrant solution for 30–45 min in a 37 °C non-CO2 incubator and the ports were loaded with a stressor mix containing 1 µM of fluoro-carbonyl cyanide phenylhydrazone (FCCP) and 1 µM of oligomycin. On the day of the assay, the culture media was replaced with Seahorse XF medium (Agilent Technologies, Santa Clara, CA, USA) supplemented with 2 mM glutamine, 1 mM pyruvate, and 10mM glucose and incubated in a 37 °C non-CO2 incubator for one hour. The energy phenotype assay was performed using the Seahorse XFe analyzer (Agilent Technologies, Santa Clara, CA, USA) as per the manufacturer’s protocol, and the results were analyzed using Wave 2.6 software (Agilent technologies, Santa Clara, CA, USA).

### 4.10. Molecular Docking and Molecular Dynamics Simulation

Molecular docking and molecular dynamics calculations were performed to analyze interactions between MSPA and its target proteins at the molecular level. Known three-dimensional structures of target proteins were obtained from the Protein Data Bank, PDB ID: 1U8F, 1YFM, 3B97, 4ALD) [38]. Models of MSPA complexed with target proteins were generated by a molecular docking calculation using Autodock Vina [39]. Molecular dynamics calculations were performed using YASARA [40] to obtain binding energies between MSPA and target proteins. Molecular dynamics simulation (at a nanosecond time-scale) of MSPA–protein complexes was performed using the AMBER force field [41] with explicit water molecules defined. The interactions between MSPA and its target proteins were analyzed using PLIP [42], and molecular visualization of the structure was performed using UCSF Chimera [43,44] and Pymol (PyMOL Molecular Graphics System, version 1.2r3pre, Schrödinger, New York, NY, USA ).

### 4.11. siRNA Sequences of Target Proteins

Four siRNAs against MSPA target genes (*ALDOA*, *GAPDH*, *ENO1*, and *FH*) were purchased as dicer-substrate short interfering duplex TriFECTa kits (Integrated DNA Technologies, Coralville, IA, USA). The sequences of siALDOA, siGAPDH, siENO1, and siFH were CGC AGG AGG AGU AUG UCA AGC GAG C, GGC CGU GAA CGA GAA GUC CUG CAA C, AGG UCG GAG UCA ACG GAU UUG GUC G, and GGA AUU UAG UGG UUA UGU UCA ACA A, respectively. Cells were transfected with the siRNAs using DharmaFECT transfection reagent (Thermo Fisher Scientific, Waltham, MA, USA) in accordance with the manufacturer’s protocol.

### 4.12. RNA Preparation and qRT-PCR Analysis

Total RNA was extracted using a Hybrid-R kit (GeneAll Biotechnology, Seoul, Korea) and reverse transcribed to cDNA using the Superscript II first-strand synthesis system (Invitrogen, Carlsbad, CA, USA). Following cDNA synthesis, PCR was performed using primers for *ALDOA*, *GAPDH*, *ENO1*, and *FH* in a dual system LightCycler (Roche Diagnostics, Rotkreuz, Switzerland), beginning with a 10 min hot start at 95 °C, followed by 45 cycles of each for 10 s at 95 °C and 30 s at 55 °C. During the last 30 s of the annealing phase, hybridization signals were measured; *HPRT1* (hypoxanthine-guanine phosphoribosyltransferase) was used to normalize gene expression. The primer and probe sequences (Roche Diagnostics, Rotkreuz, Switzerland) are listed in Appendix A.

### 4.13. Statistical Evaluation

SPSS statistics 19 (IBM Corporation, Armonk, New York, NY, USA) was used for statistical analysis. The Mann–Whitney test was used to compare cell viability, apoptosis, and enzyme activity between MSPA-treated and vehicle-treated cells. Data were expressed as the mean ± standard deviation (SD) from more than three independent experiments.

## 5. Conclusions

To the best of our knowledge, this study is the first to elucidate the mechanisms that underlie the anti-cancer properties of MSPA, which we reveal were mediated through the simultaneous suppression of key enzymes in glycolysis and the tricarboxylic acid cycle, namely *ALDOA*, *ENO1*, and *FH* (Figure 6). These enzymes are associated with the Warburg effect, a hallmark of cancer. In addition, some studies have reported that these enzymes are involved in DNA replication and response to DNA damage [35,36,45]. Thus, the inhibition of Warburg effects and the TCA cycle and the dysregulation of DNA replication with MSPA treatment could result in apoptosis. Our findings indicate that MSPA may be a promising agent for the treatment of cancer. 

## Figures and Tables

**Figure 1 pharmaceuticals-14-01060-f001:**
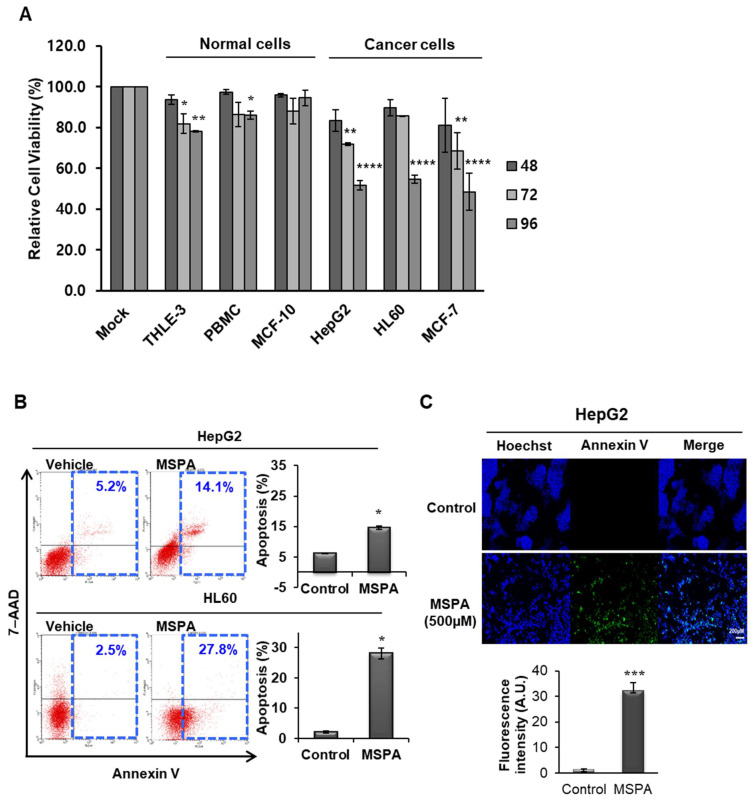
(**A**) The cytotoxic effects of macrosphelide A (MSPA) in cancer cell lines and non-cancer cell lines at 48 h, 72 h, and 96 h time points. The mock group represents vehicle-treated cell lines. Data represent the mean ± SD (*n* = 3). * *p* < 0.05, ** *p* < 0.01 **** *p* < 0.0001 versus vehicle-treated group (mock). (**B**) Apoptosis was determined using annexin V/7-aminoactinomycin D (7-ADD) staining in HepG2 and HL60 cell lines. Combined annexin V and 7-AAD reactivity allowed cells to be classified into four groups: early apoptotic cells (annexin V (+) and 7-AAD (−)), late apoptotic or dead cells (annexin V (+) and 7-AAD (+)), dead cells (annexin V (−) and 7-AAD (+)), and live cells (annexin V (−) and 7-AAD (−)). Data represent the mean ± SD (*n* = 3). * *p* < 0.05, versus vehicle-treated cells. (**C**) HepG2 cells were treated with 500 µM of MSPA for 72 h and stained with annexin V Alexa Fluor 488 (green fluorescence). Cell nuclei are stained blue (Hoechst stain). Data represent the mean ± SD (*n* = 3). *** *p* < 0.001 versus control (mock).

**Figure 2 pharmaceuticals-14-01060-f002:**
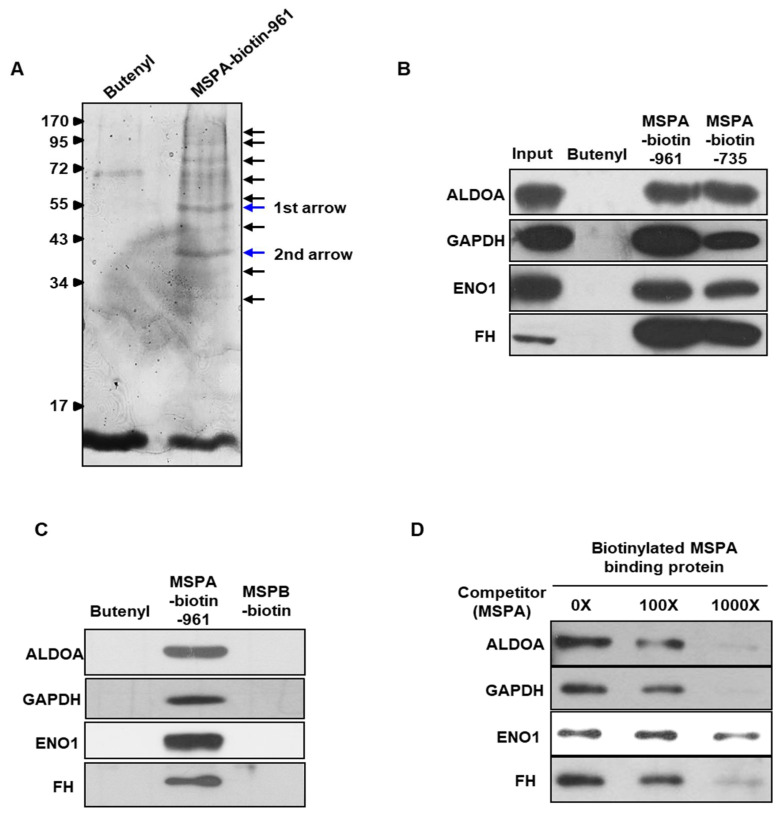
(**A**) Analysis of the putative target proteins of MSPA by Coomassie-stained SDS-PAGE analysis. Bands submitted to in-gel tryptic digestion for further mass spectrometric analysis are marked with blue arrows (first arrow and second arrow). Standard molecular weight (kDa) is on the left. (**B**) Detection of the interaction between biotinylated MSPA (or butenylated biotin, MSPA biotin-961, or MSPA biotin-735) and aldolase A (*ALDOA*), glyceraldehyde 3-phosphate dehydrogenase (*GAPDH*), enolase 1 (*ENO1*), and fumarate hydratase (*FH*) proteins. (**C**) Results of an in vitro binding assay to determine interactions between MSPA and *ALDOA*, *GAPDH*, *ENO1*, and *FH*; biotinylated MSPB was used as negative controls. (**D**) The binding specificity of MSPA in target proteins was confirmed using a competitive binding assay. Excess non-biotinylated MSPA (0× (12.5 µM), 100× (125 µM), and 1000× (1.25 mM)) was added to HL60 cell lysates as a competitor of biotinylated MSPA.

**Figure 3 pharmaceuticals-14-01060-f003:**
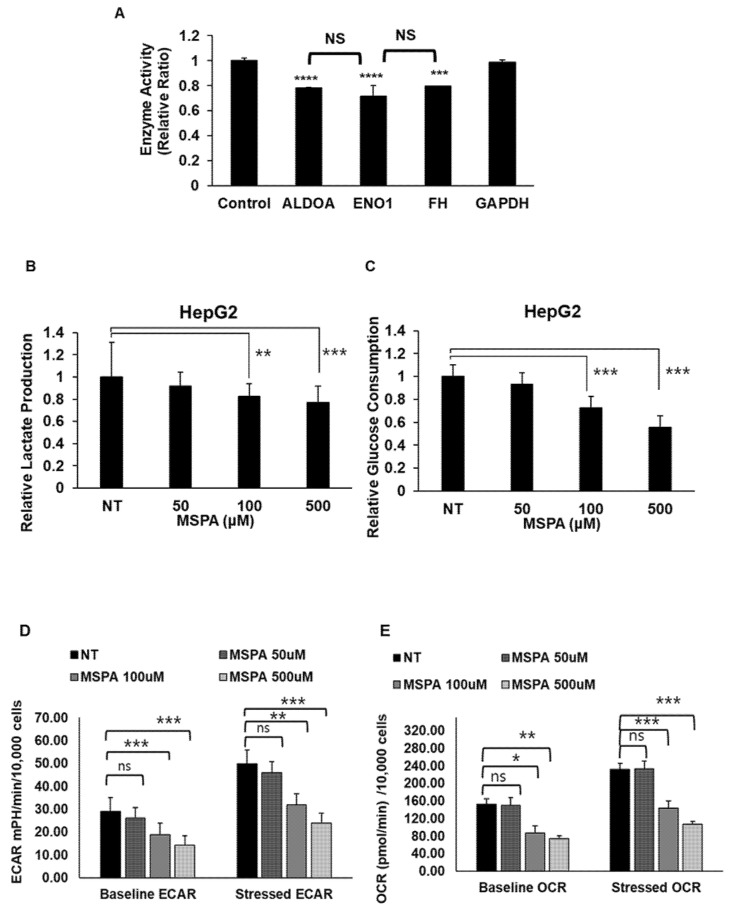
MSPA impairs glycolysis through modifying the enzymes (**A**), measurement of the effect of MSPA on the enzymatic activity of *ALDOA, ENO1*, *FH*, and *GAPDH* in HepG2 cell lines (**B**), relative lactate production (left) and (**C**), relative glucose consumption (right) in NT (DMSO-treated) and MSPA (50, 100, and 500 uM)-treated HepG2 cells for 72 h. (**D**), Extracellular acidification rate and (**E**), oxygen consumption rate in HepG2 cells treated with 50 µM, 100 µM and 500 µM of MSPA for 48 h. Data represent the mean ± SD (*n* = 3). * *p* < 0.05, ** *p* < 0.01, *** *p* < 0.001, **** *p* < 0.0001 versus the control group.

**Figure 4 pharmaceuticals-14-01060-f004:**
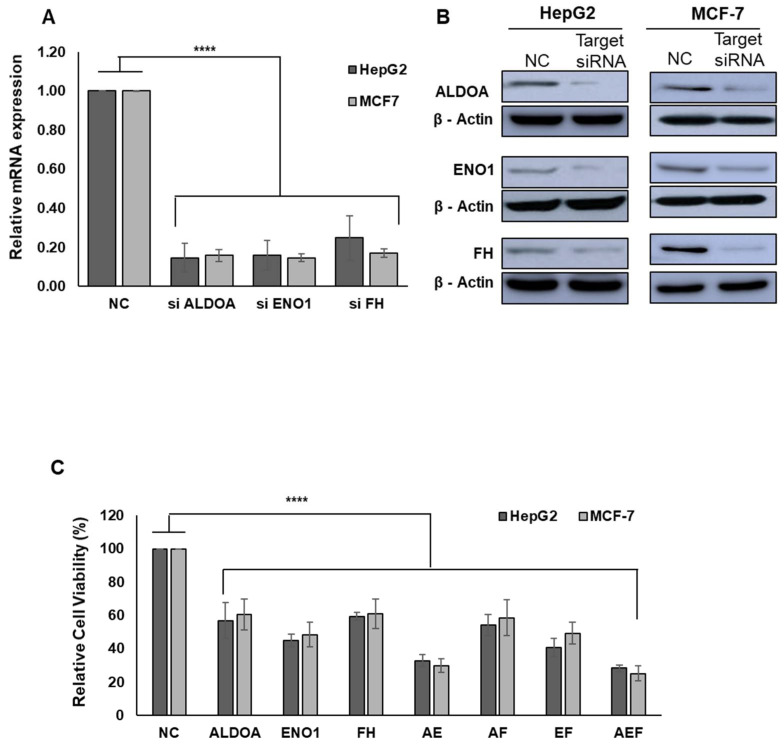
(**A**) The effect of siRNA-mediated gene knockdown on the expression level of *ALDOA*, *ENO1*, and *FH* mRNA. (**B**) The effect of *ALDOA*-, *ENO1*-, and *FH*-targeted siRNAs in HepG2 and MCF-7 cells. Lysates were prepared from cells pretreated with 1.2 nM of siRNA against a single-target enzyme. β-actin was used as a loading control. Total cellular proteins were separated using SDS-PAGE, transferred to PVDF membranes, and detected using specific antibodies against *ALDOA*, *ENO1*, *FH*, and β-actin. (**C**) The viability of HepG2 and MCF-7 cells (expressed as a percentage of the control) was determined using a CellTiter Glo luminescent assay kit. The cells were treated for 72 h with 1.2 nM of single-target siRNA (*ALDOA*, *ENO1*, or *FH*) and 1.2 nM of siRNA against more than one target (a combination of *ALDOA*, *ENO1*, and *FH*). Data were obtained from three independent experiments and presented as mean ± SD., **** *p* < 0.0001 versus the negative control siRNA (NC) group.

**Figure 5 pharmaceuticals-14-01060-f005:**
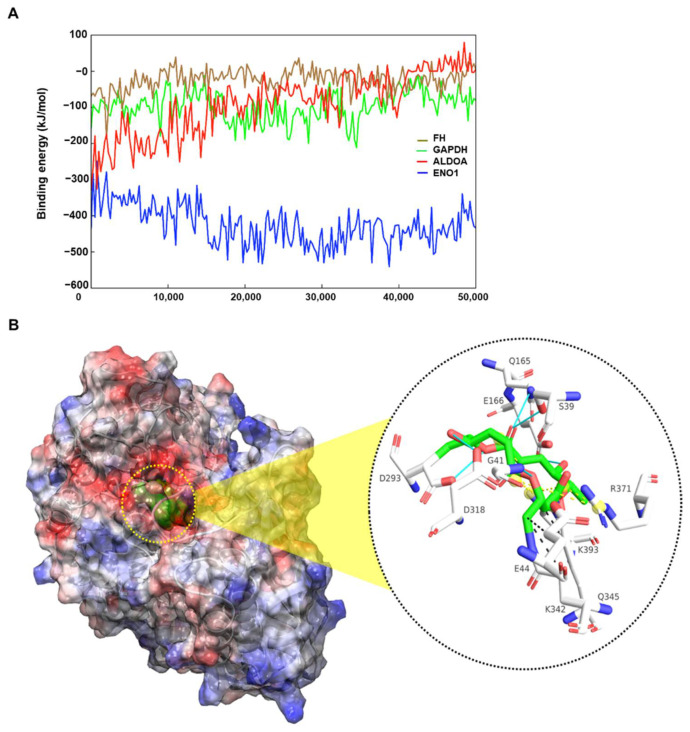
(**A**) Molecular docking calculations of MSPA–protein complexes (*GAPDH* (green), *FH* (brown), *ALDOA* (red), and *ENO1* (blue)). The graph depicts total binding energy plotted against time. (**B**) Structural representation of molecular docking analysis of MSPA in complex with the human *ENO1* protein. Models of complexed *ENO1* proteins are represented as electrostatic surface potentials (ESPs) using the ball and stick method. Positively charged surface regions are colored blue, and negatively charged surface regions are colored red. Heavy atoms are colored yellow (for sulfur), red (for oxygen), and blue (for nitrogen) in the ball and stick model of the binding site. The MSPA molecule is colored green, and interactions between proteins are represented with cyan lines (for hydrogen bonds), yellow dots (for salt bridges), and gray dots (for hydrophobic interactions).

**Figure 6 pharmaceuticals-14-01060-f006:**
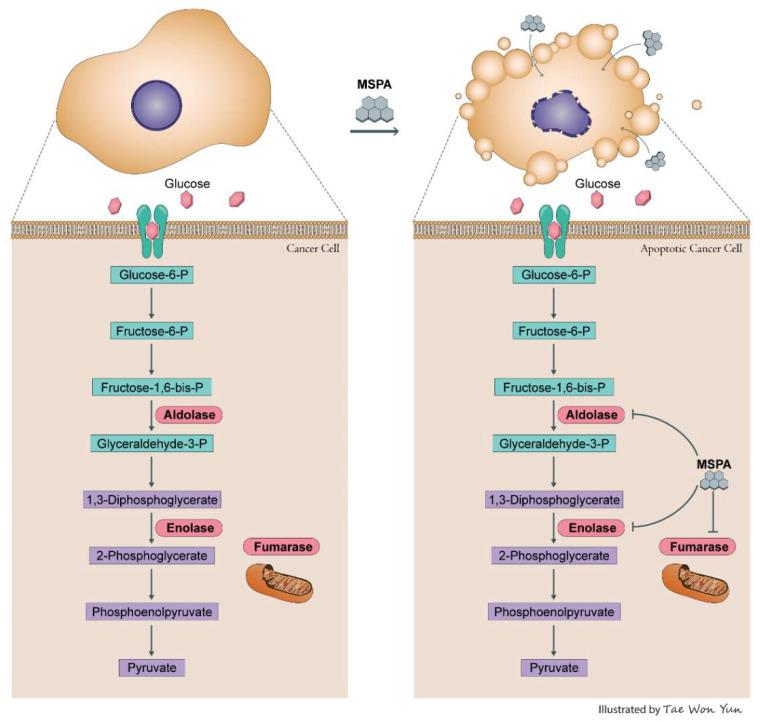
A schematic representation of the molecular mechanisms of MSPA activity; the compound inhibits cancer cell proliferation by targeting *ALDOA*, *ENO1*, and *FH*.

**Table 1 pharmaceuticals-14-01060-t001:** Identification of MSPA-interacting proteins via mass spectrometry.

Proteins Identified from around the 55 kDa Band (First Arrow in Figure 2A)			
Proteins	Peptide	Coverage (%)	Function
ENO1 (Enolase 1)	RHIADLAGNSEVILPVPAFNVINGGSHAGNK.L KLAMQEFMILPVGAANFR.E KYGKDATNVGDEGGFAPNILENKEGLELLK.T KDATNVGDEGGFAPNILENKEGLELLK.T KAGYTDKVVIGMDVAASEFFR.S KVVIGMDVAASEFFR.S RSGKYDLDFK.S RSGKYDLDFKSPDDPSR.Y KYDLDFKSPDDPSR.Y KSFIKDYPVVSIEDPFDQDDWGAWQK.F KDYPVVSIEDPFDQDDWGAWQK.F KFTASAGIQVVGDDLTVTNPK.R KFTASAGIQVVGDDLTVTNPKR.I KSCNCLLLK.V RNFRNPLAK.	43.1	Glycolytic enzyme Auto-antigen
FH (Fumarate hydratase)	RAAAEVNQDYGLDPK.I AIEMLGGELGSK.I KSKEFAQIIK.I RTHTQDAVPLTLGQEFSGYVQQVK.Y RIYELAAGGTAVGTGLNTR.I	14.9	Mitochondrial isoenzyme Involved in the TCA cycle
PDIA6 (Protein disulfide isomerase family A member 6)	RTGEAIVDAALSALR.Q KLAAVDATVNQVLASR.Y RTCEEHQLCVVAVLPHILDTGAAGR.N RGSTAPVGGGAFPTIVER.E	14.2	A member of the disulfide isomerase (PDI) family Chaperone activity
ENO2 (Enolase 2)	RAAVPSGASTGIYEALELR.D KFGANAILGVSLAVCK.A RSGETEDTFIADLVVGLCTGQIK.T	12.7	Glucose metabolism Neurotrophic and neuroprotective properties
EEF1A1 (Eukaryotic translation elongation factor 1 alpha 1)	KYYVTIIDAPGHR.D RYEEIVKEVSTYIK.K RVETGVLKPGMVVTFAPVNVTTEVK.S	11.1	Isoform of the alpha subunit of the elongation factor-1 complex Enzymatic delivery of aminoacyl tRNAs to the ribosome
**Proteins Identified from around the 40 kDa Band (Second Arrow in Figure 2A)**			
**Proteins**	**Peptide**	**Coverage (%)**	**Function**
*GAPDH* (Glyceraldehyde-3-phosphate dehydrogenase)	KVDIVAINDPFIDLNYMVYMFQYDSTHGK.F KAENGKLVINGNPITIFQER.D KLVINGNPITIFQER.D KLVINGNPITIFQERDPSK.I KWGDAGAEYVVESTGVFTTMEK.A KRVIISAPSADAPMFVMGVNHEK.Y RVIISAPSADAPMFVMGVNHEK.Y KVIHDNFGIVEGLMTTVHAITATQK.T RGALQNIIPASTGAAK.A RVPTANVSVVDLTCR.L RVVDLMAHMASK.E RVVDLMAHMASKE.	47.5	Glycolysis Uracil DNA glycosylase activity
*ALDOA* (Aldolase)	KGILAADESTGSIAK.R KIGEHTPSALAIMENANVLAR.Y RALANSLACQGKYTPSGQAGAAASESLFVSNHAY. KYTPSGQAGAAASESLFVSNHAY.	18.4	Glycolytic enzyme Gluconeogenesis
*TALDO1* (Transaldolase 1)	LIELYK.E KLSSTWEGIQAGK.E KSYEPLEDPGVK.S KIYNYYK.K KLLGELLQDNAK.L RWLHNEDQMAVEK.L	17.2	A key enzyme of the nonoxidative pentose phosphate pathway Lipid biosynthesis
*APEX1* (Apurinic/apyrimidinic endodeoxyribonuclease 1)	KGAVAEDGDELRTEPEAK.K KKNDKEAAGEGPALYEDPPDQK.T KVSYGIGDEEHDQEGR.V	16.7	Endodeoxyribonuclease activity Exonuclease activity
*ALDOC* (Aldolase)	KGVVPLAGTDGETTTQGLDGLSER.C RYASICQQNGIVPIVEPEILPDGDHDLKR.C	14.0	Glycolytic enzyme Involved in the innate immune system

## Data Availability

Data is contained within the article and Appendix A.

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
