# Peer review of "Macrosphelide A Exhibits a Specific Anti-Cancer Effect by Simultaneously Inactivating ENO1, ALDOA, and FH"

_pharmaceuticals, 2021, doi:10.3390/ph14101060_

Round 1

Reviewer 1 Report

The paper of Song et al. is devoted to study of macrosfelide A on inhibiting glycolytic enzymes – ENO1, ALDOA as well as enzyme of Kreb’s cycle FH.

The inhibition of glycolysis is very intriguing and highly desired approach to treat cancer because metabolic rewiring including increased aerobic glycolysis is well-known hallmark of various malignancies. Nevertheless, there is only a limiting progress in this direction with no compounds to be clinically approved. Thus, the research of Song et al. is dedicated to very relevant topic.

The paper is well-written and logic, no controls are missing. All approaches are relevant.

My minor comments are:

  1. Authors should add discussion based on the comparison of their results on inhibition ENO1, ALDOA and FH with literature data on inhibition of the same and other enzymes of glycolysis and Kreb’s cycle. This is very interesting to clarify the kinetic of glycolysis inhibition by several compounds. For instance, the authors have shown the really good difference of macrosfelide A-mediated inhibition of malignant and non-cancer cells at 72 and 96 hours. What suggestion do authors have to explain these results? What about literature data; does the inhibition of other glycolytic enzymes result in the same rather late manifestation of the effect (72h, 96h or late)?
  2. The authors have shown macrosfelide A-mediated inhibition of enzymatic ENO1, ALDOA and FH activity. These data mean the inhibition of glycolysis (ENO1 and ALDOA) and respiration (all three enzymes). Thus, the study of macrosfedile A direct influence on glycolysis and respiration is highly required. For instance, to simplify this research, the SeaHorse XF Energy Phenotype Kit can be used which simultaneously detects both glycolysis (ECAR) and respiration (OCR), as well as stressed ECAR and OCR. Does the significant inhibition of glycolysis and respiration in cancer cells manifest only on 72 and 96 hours?

Reviewer 2 Report

This manuscript by Song et al. is quite relevant and of interest since tumor metabolism represents a hot topic. Here are my comments:

  • Seahorse experiments should be done to confirm the observed results.
  • Add these 2 seminal references: 
    • Cassim S, Raymond VA, Dehbidi-Assadzadeh L, Lapierre P, Bilodeau M. Metabolic reprogramming enables hepatocarcinoma cells to efficiently adapt and survive to a nutrient-restricted microenvironment. Cell Cycle. 2018;17(7):903-916. doi: 10.1080/15384101.2018.1460023. Epub 2018 May 21. PMID: 29633904; PMCID: PMC6056217.
    • In vivo experiments should be discussed in the Discussion part - 1/2 sentences.

Author Response

This manuscript is a resubmission of an earlier submission. The following is a list of the peer review reports and author responses from that submission.

Round 1

Reviewer 1 Report

The authors investigate the effect of Macrosphelide A on cancer cells and normal cells culture in vitro. The authors conduct binding assays and identify ALDOA, ENO1, FH and GAPDH as proteins binding to Macrosphelide A. They also conducted enzymatic assays to measure the activities of these enzymes, in control and Macrosphelide A treatment. The authors also perfom in silico calculations of Macrosphelide A binding to ALDOA, ENO1, FH and GAPDH, suggesting that binding is stronger to ENO1, which appears to exhibit the strongest enzymatic inhibition by Macrosphelide A.

Comments

1- The in silico analysis indicates stronger biding of Macrosphelide A to ENO1 than to the other enzymes. However, it is not obvious form figure 3 that the inhibition of ENO1 enzymatic activity is significantly stronger than ALDOA and FH. The authors should report the statistical significance of differences between the enzymatic activity of ENO1 vs ALDOA and ENO1 vs FH.

2- The reduction of enzymatic activity of ENO1, ALDOA and FH are quite modest (Fig. 3). It is not evident from this data that Macrosphelide A inhibits the rate of the associated enzymes in cells. The authors should conduct experiments to determine if Macrosphelide A inhibits glycolysis (ALDOA, ENO1) and the TCA cycle (FH) in HepG2 cells. Measurement of glucose uptake and lactate release could be use as surrogates of the rate of glycolysis. Incorporation of 13C from glucose or glutamine into TCA cycle metabolites could be used to estimate the TCA cycle activity.

Reviewer 2 Report

In this submitted manuscript, Song and colleagues have identified a cancer cell-selective toxic property of a compound they have syntheized, MSPA. Using a biotin pulldown approach they identify in an unbiased manner several glycolysis enzymes as MSPA binders. Using molecular docking calculations and molecular simulations, they provide structural rationale for how MSPA might be binding and inhibiting these enzymes.

As the authors discuss, ‘Warburg effect’ and dependency on glycolysis is an attractive target for cancer therapy, and the authors provide compelling and clear data, in particular the pulldown data which is very unambiguous. Functional characterization of the effects of MSPA on cancer vs non-cancer cells is mostly sound. The manuscript is very well written and of sufficient impact and interest to the readership. However, I do have one major issue that I feel should be addressed for the study to be suitable for final publication. The enzyme activity assays provided in Figure are from ‘kits’ that measure various readouts from cell lysates. These can be affected by many indirect/secondary factors and are thus far inferior to a real activity assay with recombinant enzyme expressed and purified from bacteria, which is the ideal way to demonstrate direct inhibition of enzyme activity. If the authors can carry out such an approach for at least one of the several enzymes they have shown to bind MSPA, as a reviewer I would be satisfied and find their study to be very convincing.

As a minor point, the annexin V staining shown in Figure 1C is strange – Annexin V should stain the cell periphery, not the nucleus. Also, the authors have excessively zoomed into a single nucleus which is neither informative nor convincing. The authors should revisit this experiment, provide correct annexin V staining data showing multiple cells in the micrograph field, and ideally provide a blinded count data.
